# Oxidative Stress and Risk of Dementia in Older Patients with Depression: A Longitudinal Cohort Study Using Plasma Biomarkers

**DOI:** 10.3390/medicina61010108

**Published:** 2025-01-13

**Authors:** Yoo-Jin Jang, Min-Ji Kim, Su-Jin Lee, Shinn-Won Lim, Doh-Kwan Kim

**Affiliations:** 1Department of Psychiatry, Samsung Medical Center, Sungkyunkwan University School of Medicine, 81 Irwon-ro, Gangnam-gu, Seoul 06351, Republic of Korea; yjjjs519@naver.com; 2Biomedical Statistics Center, Research Institute for Future Medicine, Seoul 06355, Republic of Korea; rabbit93.kim@samsung.com; 3Department of Health Sciences and Technology, Samsung Advanced Institute for Health Sciences & Technology, Sungkyunkwan University, Seoul 06351, Republic of Korea; lovegun0311@naver.com

**Keywords:** late-life depression, Alzheimer’s dementia, oxidative stress, biomarker, nitrotyrosine

## Abstract

*Background and Objectives*: While depression is associated with an increased risk of Alzheimer’s dementia (AD), traditional AD-related biomarkers, such as amyloid-beta, have shown limited predictive value for late-life depression. Oxidative stress has emerged as a potential indicator given its shared role in both depression and dementia. This study investigated the longitudinal relationship between oxidative stress biomarkers and risk of dementia in patients with depression. *Materials and Methods*: A longitudinal cohort of 146 older patients with major depressive disorder was analyzed. Biomarkers, such as nitrotyrosine, protein carbonyl, F2-isoprostanes, malondialdehyde, 4-hydroxynonenal, and 8-hydroxy-2′-deoxyguanosine, were collected at baseline and measured using an enzyme-linked immunosorbent assay. AD conversion was determined using comprehensive neuropsychological assessment. Cox proportional hazards models were used to evaluate the association between oxidative stress biomarkers and AD conversion after adjusting for confounders. The log-rank test, using the minimum *p*-value approach, was applied to determine the optimal cut-off value for significantly associated biomarkers of AD-free survival rates. *Results:* During the follow-up period ranging from 1.00 to 18.53 years, 41 (28.08%) patients converted to AD. Nitrotyrosine showed a significant association with increased risk of AD (adjusted hazard ratio [HR], 1.01; 95% confidence interval [CI], 1.00–1.01; *p* = 0.0045). For clinical applicability, patients with plasma nitrotyrosine levels ≥170 nM as the cut-off value had a 5.14-fold increased risk of AD (adjusted HR, 5.14; 95% CI, 2.02–13.07; *p* = 0.0006). Other biomarkers, such as protein carbonyl, F2-isoprostanes, malondialdehyde, 4-hydroxynonenal, and 8-hydroxy-2′-deoxyguanosine, were not significantly associated with AD conversion. *Conclusions:* Nitrotyrosine, a biomarker that reflects nitrosative damage, emerged as a significant predictor of dementia risk in older patients with depression, highlighting its potential as an early biomarker of dementia. Further validation of these results is required using a larger sample size.

## 1. Introduction

Depression is associated with an increased risk of Alzheimer’s dementia (AD) [1]. Meta-analyses and longitudinal studies have reported that older adults with depression have a significantly elevated risk of developing dementia, with hazard ratios ranging from 1.85 to 2.83 [2,3]. This strong association highlights the importance of identifying predictive biomarkers that can help detect early dementia risk in high-risk populations. However, the evidence linking traditional Alzheimer’s disease-related biomarkers, such as amyloid-beta (Aβ), to late-life depression and the risk of dementia in patients, remains unclear [4,5]. This suggests that the biomarkers used to predict dementia in the high-risk group may differ from those identified in the general population. Identifying specific predictive biomarkers for individuals with depression is crucial for early detection and intervention, enabling personalized strategies to mitigate the progression of dementia.

Oxidative stress, a common pathological factor in depression and dementia, may be a potential indicator [6,7,8]. Increased oxidative stress and reduced antioxidant capacity have been observed in patients with depression [6]. Reactive oxygen species (ROS) play vital roles in normal cell signaling and defense processes. However, excessive ROS production induced by depression can trigger inflammatory cascades, damage key cellular components, and promote apoptosis, leading to cognitive decline [9]. The brain is particularly vulnerable to oxidative stress owing to its high oxygen consumption, abundant lipid content, and limited antioxidant defenses [10].

Previous studies have extensively examined the role of oxidative stress in dementia and have shown significantly higher levels of oxidative stress biomarkers in patients with Alzheimer’s disease than in controls [11]. However, these investigations have largely focused on the general population or on individuals already diagnosed with dementia [12], leaving a critical gap in the understanding of how oxidative stress contributes to the risk of dementia in individuals with depression. Furthermore, most previous research has been cross-sectional [13], limiting insights into the longitudinal association between oxidative stress biomarkers and the risk of dementia [14]. While previous studies [15] have explored the longitudinal relationship between oxidative stress and dementia progression, their follow-up duration was relatively short, ranging from 1 to 6 years. Given the growing evidence that depression is a prodrome or risk factor for dementia, it is imperative to identify specific biomarkers that can predict dementia in patients with depression.

To address this gap, we hypothesize that oxidative stress biomarkers are significantly associated with an increased risk of dementia subsequently developing in individuals with depression. Specifically, we propose that elevated levels of oxidative stress markers, such as ROS, and their downstream effects may serve as early indicators of dementia risk. By investigating the longitudinal relationship between oxidative stress biomarkers and dementia in a cohort of individuals with depression, this study aimed to clarify the predictive utility of these biomarkers and provide a foundation for personalized prevention strategies.

## 2. Materials and Methods

### 2.1. Study Cohort and Participant Selection

This study was based on a retrospective analysis of data from a larger cohort initially established for research on antidepressant treatment responses in a naturalistic clinical setting [16]. Our cohort of individuals with depression consisted of those who met the Diagnostic and Statistical Manual of Mental Disorders, Fourth Edition (DSM-IV) criteria for major depressive disorder [17] and had a baseline 17-item Hamilton Depression Rating Scale (HAM-D) score of at least 15 [18]. At baseline, patients were excluded if they had psychotic disorders (such as schizophrenia or delusional disorder), bipolar affective disorder, neurological illnesses (such as Parkinson’s disease and epilepsy), intellectual developmental disability, significant medical conditions, history of alcohol or drug dependence, personality disorders, head trauma with loss of consciousness, malignancy, or abnormal baseline laboratory findings, to focus on unipolar depression and avoid the potential confounding effects of these comorbidities. Furthermore, patients with unstable psychiatric conditions posing immediate risks or challenges to stable follow-up (e.g., recent suicide attempts during a depressive episode) were also excluded. All participants were recruited from a geropsychiatric clinic at Samsung Medical Center between June 1998 and January 2012.

This cohort comprised patients who initially sought treatment for depression and continued to receive care at the Samsung Medical Center. This naturalistic setting allowed many participants to be followed up consistently, even beyond the initial studies on antidepressant treatment responses, facilitating the long-term tracking of clinical and biological changes. Over time, the cohort was expanded and followed up longitudinally to explore various aspects of late-life depression, including clinical, genetic, pharmacokinetic, and immune factors. This robust dataset provided a unique opportunity to investigate the interplay between depression and dementia.

From this cohort, we identified 146 patients aged ≥ 55 years who were cognitively normal at baseline (Korean Mini-Mental State Examination [K-MMSE] score ≥ 28/30 [19]) and had available plasma samples for oxidative stress biomarker analysis. This subset allowed us to investigate the relationship between oxidative stress biomarkers and risk of dementia in cognitively healthy individuals with depression.

### 2.2. Study Protocol

This study was approved by the ethics review board of Samsung Medical Center (IRB approval no. 2006-03-012). Written informed consent was obtained from all the participants.

At enrollment, all participants underwent a structured research interview using the Samsung Psychiatric Evaluation Schedule (SPES) [20]. The SPES collects data on psychiatric symptoms, cognitive screening, comorbid physical diagnoses (hypertension, diabetes mellitus, dyslipidemia, cardiac disease, and cerebrovascular disease), and psychosocial variables (age, sex, education, age at the onset of depression, duration of the current episode, number of depressive episodes, family history of depression, and initial HAM-D score). Each diagnostic interview involved the patient and at least one family member. A board-certified psychiatrist with extensive clinical experience in geriatric psychiatry confirmed all diagnoses using SPES, clinical observations, and medical records. Peripheral blood samples were collected at baseline for routine apolipoprotein E (ApoE) genotyping and biomarker analyses.

Participants were followed up every 3 months until 21 December 2021, onset of dementia, patient death, or lost to follow-up. Antidepressant treatment response was categorized as a response or remission based on the HAM-D score at 6 weeks [21]. Response was defined as a ≥50% reduction in the HAM-D score, while remission was defined as a HAM-D score ≤ 7 after 6 weeks of treatment. The K-MMSE, a validated and reliable tool for cognitive screening in Korean-speaking populations [19], was updated annually. If cognitive decline was reported by the patient, caregiver, or clinician, additional neuropsychological assessments, brain magnetic resonance imaging (MRI), and laboratory tests were performed. Neuropsychological assessments, including the K-MMSE, Clinical Dementia Rating (CDR) scale [22], Seoul Neuropsychological Screening Battery-Dementia version [23], Seoul-Activities of Daily Living [23], Seoul-Instrumental Activities of Daily Living [24], Korean version of the Neuropsychiatric Inventory [25], and Korean version of the Geriatric Depression Scale [26] were also performed. These validated cognitive assessments were conducted by clinical psychologists trained in the standardized administration of these tools to ensure consistency and reliability of data collection. Brain MRI findings were reviewed by board-certified neuroradiologists and used as a supplementary tool to distinguish between other conditions that could lead to dementia syndrome.

### 2.3. Biomarkers of Oxidative Stress

To comprehensively assess the oxidative damage, we focused on six key biomarkers established in previous studies. Nitrotyrosine is formed when tyrosine residues in proteins are nitrated by reactive nitrogen species (RNS), reflecting protein modifications under oxidative stress [27]. Protein carbonyl is a well-established biomarker of protein oxidation, indicating damage to protein structure [28]. For lipid peroxidation, we used F2-isoprostanes as precise indicators of oxidative damage to the cell membranes [29]. 4-hydroxynonenal (4-HNE) is a reactive aldehyde produced primarily by the oxidation of unsaturated fatty acids in cell membranes [30], whereas malondialdehyde (MDA) is another commonly measured end product of lipid peroxidation [31]. Lastly, 8-hydroxy-2′-deoxyguanosine (8-OHdG) is used as a representative marker of oxidative damage to DNA [32]. These six biomarkers were selected based on their consistent identification in previous studies as key indicators of oxidative stress linked to neurodegeneration and dementia, emphasizing their clinical relevance and suitability for measurements using established methodologies.

Peripheral venous blood samples (10 mL) were collected from patients with depression using EDTA K2 anticoagulant vacutainer tubes (Becton Dickinson, Madrid, Spain) between 8:00 AM and 10:00 AM at baseline. Plasma was separated by centrifugation at 3000 rpm for 10 min, aliquoted, and stored at −80 °C until analysis. Plasma level of biomarkers related to oxidative stress was measured using an enzyme-linked immunosorbent assay kit (Abcam, Cambridge, UK for nitrotyrosine, protein carbonyl, and 8-OHdG; MyBioSource, San Diego, CA, USA for F2-isoprostanes; Cell Biolabs, Inc., San Diego, USA for 4-HNE and MDA) for the analysis of nitrotyrosine, protein carbonyl, 8-OHdG, F2-isoprostanes, 4-HNE, and MDA according to the manufacturers’ instructions. Each sample was tested in duplicate, and standards were included in each plate to create a standard curve. The intra- and inter-assay coefficients of variation (intra-/inter-assay CV, %) for assayed measures of nitrotyrosine, protein carbonyl, 8-OHdG, F2-isoprostanes, 4-HNE, and MDA in two samples were 2.43/4.42, 5.67/3.77, 3.85/1.75, 5.26/3.95, 3.94/3.82, and 2.63/2.38, respectively. A small proportion of samples, 2.74% for F2-isoprostanes and 8.22% for 4-HNE, had cytokine levels below the detection limit (<0.01 pg/mL) and were indistinguishable from background noise. These samples were subsequently assigned a value of ‘zero’ for the respective cytokine.

### 2.4. Conversion to AD

The primary outcome was conversion to AD as defined by the DSM-IV. Participants with a CDR score ≥ 1 had their diagnoses confirmed by a clinician using the DSM-IV criteria, neuropsychological testing, and impairments in activities of daily living. Probable AD was diagnosed using the National Institute of Neurological and Communicative Diseases and the Stroke-Alzheimer’s Disease and Related Disorders Association criteria [33]. Participants newly diagnosed with dementia underwent annual follow-up neuropsychological tests to confirm their status and dementia subtype. The date of onset of dementia was recorded as the first confirmed diagnosis. For diagnostic accuracy, individuals exhibiting signs of non-AD neurodegenerative disorders, such as Parkinsonian features or with pronounced behavioral and personality alterations, were not classified as having had AD conversion. The diagnostic consistency between observers in distinguishing AD from non-AD cases in our geropsychiatric clinic was 91.4%.

### 2.5. Statistical Analysis

All statistical analyses were performed using SAS version 9.4 (SAS Institute, Cary, NC, USA), with the significance level set at *p* < 0.05. The clinical and demographic characteristics of the participants are presented as categorical or continuous variables, where categorical variables are expressed as frequencies and proportions, and continuous variables as mean ± standard deviation. Data distribution and normality were assessed using box plots, Q-Q plots, and Shapiro–Wilk tests. The log10 transformation was applied to biomarkers with highly skewed distributions. Univariable Cox proportional hazards regression was conducted to determine the hazard ratios (HRs) and 95% confidence intervals (CIs) of the clinical and demographic variables for AD conversion after verifying the proportional hazard assumption using Schoenfeld residuals. Multivariable Cox regression was performed, adjusting for variables with *p* < 0.1 in the univariable analysis. Collinearity was evaluated using the variance inflation factor to ensure independent contributions of the predictors. For biomarkers with a significant association, the cutoff value was determined using the minimum *p*-value approach from the log-rank test of AD-free survival rates.

## 3. Results

### 3.1. Participant Characteristics

Table 1 presents the baseline clinical and demographic characteristics of the participants along with the results of the univariable Cox proportional hazards regression. Among the 146 participants, 41 (28.08%) converted to AD during the follow-up period ranging from 1.00 to 18.53 years, and median follow-up time (95% CI) was 6.2 (4.8–9.2) years estimated by reverse Kaplan–Meier method. Among the remaining 105 participants, 34 died, 55 were lost to follow-up, and 16 completed the follow-up period by 31 December 2021.

Older age was associated with an increased risk of AD (HR, 1.06; 95% CI, 1.01–1.11; *p* = 0.0266). Lower education levels also increased the risk of AD (HR, 0.93; 95% CI, 0.87–0.99; *p* = 0.0314). Participants with depression onset at age ≥ 75 years had a markedly higher risk of AD conversion compared to those with earlier onset (HR, 3.78; 95% CI, 1.81–7.91; *p* = 0.0004). A longer duration of the current depressive episode was associated with an increased risk of AD (HR, 1.03; 95% CI, 1.00–1.05; *p* = 0.0230), while a higher number of depressive episodes was inversely associated with AD risk (HR, 0.68; 95% CI, 0.49–0.93; *p* = 0.0162). Regarding comorbidities, cardiac disease (HR, 3.72; 95% CI, 1.53–9.00; *p* = 0.0036) and cerebrovascular disease (HR, 2.24; 95% CI, 1.09–4.61; *p* = 0.0282) were significantly associated with an increased risk of AD conversion.

Seven covariates (age, education, depression onset age ≥ 75 years, number of depressive episodes, duration of the current episode, cardiac disease, and cerebrovascular disease) at a *p* < 0.1 in the univariable analysis (Table 1) were included as covariates in the multivariable analysis.

### 3.2. Association of Oxidative Stress Biomarkers with AD Conversion

Table 2 shows the plasma levels of oxidative stress biomarkers and their association with AD conversion. Among the oxidative stress biomarkers examined, nitrotyrosine was significantly associated with an increased risk of developing AD. In the univariable analyses, higher levels of nitrotyrosine were associated with an HR of 1.01 (95% CI, 1.00–1.01; *p* = 0.0090). Protein carbonyl showed a trend toward significance (HR, 2.92; 95% CI, 0.94–9.11; *p* = 0.0652), while other biomarkers, including F2-isoprostanes, 4-HNE, MDA, and 8-OHdG, did not show significant associations with AD conversion. In a multivariable model that included both nitrotyrosine and protein carbonyl, along with covariates (age, education, depression onset age ≥ 75 years, number of depressive episodes, duration of the current episode, cardiac disease, and cerebrovascular disease), higher levels of nitrotyrosine remained significantly associated with an increased risk of AD conversion (adjusted HR, 1.01; 95% CI, 1.00–1.01; *p* = 0.0045). However, the protein carbonyl levels were not significantly associated with AD conversion in the same model.

### 3.3. Clinical Applicability of Candidate Oxidative Stress Markers

For clinical applicability, we identified the optimal cutoff value for nitrotyrosine in the prediction of AD, with 170 nM being explored as a potential threshold. At a cutoff point of 170, eighteen participants had nitrotyrosine levels above this threshold, and seven converted to AD. The log-rank test showed a significant difference in AD-free survival curves between participants with nitrotyrosine levels ≥ 170 nM and those with levels < 170 nM (*p* = 0.0030). Cox proportional hazards analysis adjusted for covariates, including age, education, depression onset age ≥ 75 years, number of depressive episodes, duration of the current episode, cardiac disease, and cerebrovascular disease, further confirmed this association. Participants with nitrotyrosine levels ≥ 170 nM had significantly lower AD-free survival rates than those with nitrotyrosine levels < 170 nM. Over the follow-up period, the adjusted HR for AD conversion in the ≥170 nM group was 5.14 (95% CI, 2.02–13.07; *p* = 0.0006, Figure 1).

## 4. Discussion

In this longitudinal cohort study of oxidative stress biomarkers, we highlight the role of nitrotyrosine as a potential biomarker for predicting AD conversion in individuals with depression. In terms of clinical applicability, patients with plasma nitrotyrosine levels ≥ 170 nM had a 5.14-fold increased risk of AD. No significant associations were observed with other markers of oxidative stress.

Our findings align with those of previous studies that have shown associations between nitrotyrosine and dementia in populations without depression [13,34,35]. Importantly, nitrotyrosine levels predicted the risk of dementia in individuals with depression with a significant longitudinal association (Table 2). Given that evidence linking amyloid burden to dementia development in depressed populations remains inconclusive [4,5], oxidative stress could be an alternative pathway leading to dementia in this vulnerable group. Nitrotyrosine, a protein nitration marker, impairs neuronal function and accelerates neurodegenerative changes over time [27]. Our findings suggest that plasma nitrotyrosine levels may serve as an early indicator of brain damage, even in cognitively normal individuals with depression, highlighting their potential as a biomarker for predicting the risk of dementia. We suggest a candidate cutoff value of 170 nM for nitrotyrosine. Participants with nitrotyrosine levels ≥ 170 nM had a 5.14-fold higher risk of developing AD than those with lower levels (Figure 1). There are studies that did not find a significant association between nitrotyrosine levels and dementia. For example, one study reported no differences in plasma nitrotyrosine concentrations between patients with AD and age-matched controls [36]. This discrepancy may arise due to differences in study populations and methodological approaches used to measure the biomarkers. Additionally, the smaller sample size of that study (22 patients with AD and 18 age-matched controls) may have limited the power to detect significant associations. Further validation with larger sample sizes is essential to confirm the utility of nitrotyrosine in predicting dementia and to establish an optimal cutoff value for clinical use.

Other oxidative stress markers, such as F2-isoprostanes, 4-HNE, MDA, and 8-OHdG, did not show statistically significant associations, which contrasts with findings of previous general population-based studies [13,14,30,32]. This discrepancy could be attributed to several factors. First, differences in the biomarker stability and sensitivity of the assays used might have contributed to the variability in the measurements. Lipid peroxidation markers, such as MDA and 4-HNE, are highly reactive and may degrade during sample storage or handling, potentially affecting the reliability of their associations [37]. Second, the biological conditions of the individuals with depression might have influenced the results. Depression is often associated with chronic activation of the hypothalamic–pituitary–adrenal axis, which can lead to reduced antioxidant capacity and accumulation of oxidative stress [6]. This potential increase in oxidative stress levels among patients with depression may homogenize the biomarker levels within the cohort, thus reducing the ability to detect meaningful differences between individuals who develop AD and those who do not.

However, these factors do not fully explain why nitrotyrosine alone showed a statistically significant association in our results. The unique pathological properties of nitrotyrosine may provide a plausible explanation. Unlike other markers that primarily reflect lipid peroxidation or DNA damage at the subcellular or intracellular levels, nitrotyrosine causes functional modifications in proteins, potentially exerting greater toxicity at the cellular level. Consequently, it can cause more severe damage than the other oxidative markers. Although not statistically significant, the relatively high HR for protein carbonyl further supports the role of protein oxidation in neurodegenerative processes (Table 2). Furthermore, the lack of significant association with other markers may reflect the secondary role of oxidative stress in the pathophysiology of depression-associated dementia. Nitrotyrosine specifically reflects the oxidative damage associated with RNS, which is closely related to both oxidative stress and inflammatory responses. When inflammatory cells such as microglia are activated, along with the release of pro-inflammatory cytokines such as interleukin-1 beta (IL-1β) and tumor necrosis factor-α, large amounts of nitric oxide and RNS are produced [38]. This leads to the formation of peroxynitrite, which promotes the generation of protein nitration, resulting in a combination of oxidative and inflammatory damage. We previously identified a significant association between IL-1β and AD conversion in patients with depression, suggesting the neuroinflammatory hypothesis linking depression with AD [39]. Oxidative stress may not be the primary contributor to neurodegeneration, but rather a by-product of inflammatory pathways that accompany and amplify the damage initiated by inflammation. Although our findings indicate that the utility of nitrotyrosine in predicting dementia in patients with depression is promising, other oxidative stress markers may not be directly applicable to this population.

Our findings provide a foundation for future research on the complex interplay between oxidative stress, inflammation, and neurodegeneration. Potential research might involve an investigation of the gut–brain axis, as recent studies have highlighted the influence of the gut microbiome on oxidative stress and inflammatory pathways [40,41,42]. Alterations in the composition of gut microbiota and their metabolites exacerbate systemic oxidative stress and inflammation, which are key contributors to depression and dementia. Integrating microbiome profiling with oxidative stress biomarkers may enhance predictive models and inform microbiome-targeted therapies to mitigate neuroinflammation and oxidative damage. Such research may lead to novel personalized approaches to prevent or delay dementia in at-risk populations.

This study has several limitations. First, the absence of a healthy control group limited the ability to confirm a specific association between oxidative stress and dementia. Our primary aim was to identify biomarkers relevant to patients with depression transitioning to dementia, which may differ from those in the general population. Second, as this study retrospectively analyzed data from a cohort originally established for other purposes, we were unable to calculate an a priori sample size tailored to our specific research aims. Our post hoc calculation showed that the power for the adjusted HR of 1.01 for nitrotyrosine was 89%, indicating sufficient statistical power, given an observed *p*-value of 0.0045. While post hoc power analyses are inherently limited and do not replace pre-study power calculations [43], this result indicates that our study had sufficient power to detect observed associations, thereby supporting the robustness of our findings. Third, the substantial dropout rate over the nearly 20-year follow-up period may have affected the internal validity of the study and reduced its statistical power. Fourth, the use of long-storage plasma samples may have affected the measurement accuracy. Nevertheless, oxidative stress biomarkers have shown good stability even during long-term storage [44]. To enhance reliability, rigorous quality controls, including duplicate measurements and intra- and inter-assay consistency checks, were used to confirm the reliability of the results (Methods). Fifth, our strict cognitive criteria (K-MMSE ≥ 28) may limit generalizability, as late-life depression often involves some cognitive impairment. However, focusing on cognitively intact patients provides information on early dementia biomarkers. Sixth, peripheral oxidative stress markers provide an indirect measure that may not capture the localized oxidative processes occurring within the brain. Lastly, while we adjusted for various covariates in the analysis, other potential confounding factors, such as dietary habits, smoking status, and medication use, were not accounted for. These factors are known to influence oxidative stress biomarker levels and may have introduced bias into our findings.

## 5. Conclusions

This study showed that nitrotyrosine, a biomarker of RNS-associated oxidative damage, was significantly associated with an increased risk of AD in patients with depression. These findings highlight the potential of nitrotyrosine as an early predictor of AD, even in cognitively normal patients, as well as its role in oxidative and inflammatory damage during neurodegeneration. Further validation studies with larger cohorts are needed to confirm the optimal cutoff value and establish the utility of nitrotyrosine as a biomarker in clinical practice.

## Figures and Tables

**Figure 1 medicina-61-00108-f001:**
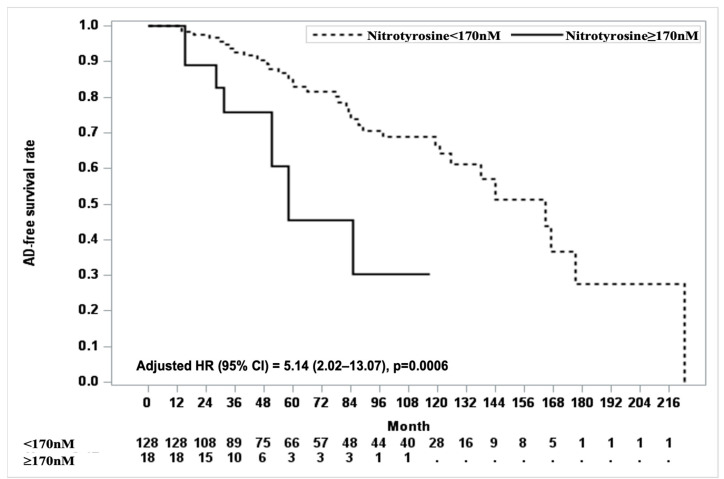
Kaplan–Meier curves for AD-free survival based on nitrotyrosine cut-off value. The Kaplan–Meier curve shows AD-free survival rates based on nitrotyrosine levels with a cut-off value of 170. Participants with nitrotyrosine levels < 170 nM (dashed line) are compared to those with levels ≥ 170 nM (solid line). The x-axis represents the follow-up duration in months, while the y-axis represents the proportion of participants who remain AD-free. The number of participants at risk for each group (below and above the cut-off points) is displayed at the bottom of the figure.

**Table 1 medicina-61-00108-t001:** Clinical and demographic characteristics of participants.

	Total	Conversion to AD		Hazard Ratio	*p*-Value
	(*n* = 146)	No (*n* = 105)	Yes (*n* = 41)	(95% CI)	
Sex (female)	123 (84.25)	88 (83.81)	35 (85.37)	1.02 (0.42–2.46)	0.9706
Age (years)	69.66 ± 7.38	69.40 ± 7.34	70.32 ± 7.55	1.06 (1.01–1.11)	0.0266
Education (years)	8.65 ± 4.77	9.06 ± 4.63	7.61 ± 5.03	0.93 (0.87–0.99)	0.0314
Onset age ≥ 75 years	21 (14.38)	11 (10.48)	10 (24.39)	3.78 (1.81–7.91)	0.0004
Initial HAM-D	19.18 ± 4.36	19.33 ± 4.18	18.66 ± 4.94	0.93 (0.85–1.02)	0.1198
Response	100 (68.49)	69 (65.71)	31 (75.61)	1.67 (0.81–3.43)	0.1618
Remission	60 (41.10)	40 (38.10)	20 (48.78)	1.36 (0.73–2.54)	0.3321
Family history of depression	22 (15.07)	19 (18.10)	3 (7.32)	0.61 (0.19–1.99)	0.4117
Number of depressive episodes	2.29 ± 1.94	2.48 ± 2.14	1.80 ± 1.14	0.68 (0.49–0.93)	0.0162
Duration of current episode (months)	8.71 ± 10.56	7.00 ± 7.06	13.10 ± 15.72	1.03 (1.00–1.05)	0.0230
Comorbidities					
Hypertension	57 (39.04)	37 (35.24)	20 (48.78)	1.54 (0.82–2.89)	0.1760
Diabetes mellitus	23 (15.75)	18 (17.14)	5 (12.20)	0.78 (0.31–1.99)	0.6026
Dyslipidemia	15 (10.27)	11 (10.48)	4 (9.76)	0.99 (0.35–2.81)	0.9906
Cardiac disease	11 (7.53)	5 (4.76)	6 (14.63)	3.72 (1.53–9.00)	0.0036
Cerebrovascular disease	18 (12.33)	8 (7.62)	10 (24.39)	2.24 (1.09–4.61)	0.0282
*ApoE4* allele, 0	114 (78.08)	85 (80.95)	29 (70.73)		
*ApoE4* allele, 1 or 2	32 (21.92)	20 (19.05)	12 (29.27)	1.38 (0.68–2.77)	0.3726

HAM-D, 17-item Hamilton Rating Scale for Depression; AD, Alzheimer’s dementia; ApoE4, apolipoprotein E4; CI, confidence interval. Data are presented as mean ± standard deviation for continuous variables and as frequency (percentage) for categorical variables. Cox proportional hazards regression analysis was performed to determine the association between demographic and clinical factors and conversion to AD.

**Table 2 medicina-61-00108-t002:** Plasma levels of oxidative stress and association of oxidative stress markers with Alzheimer’s dementia.

		Conversion to AD		Univariable Analysis		Multivariable Analysis *	
Oxidative Stress Markers	Total (*n* = 132)	No (*n* = 98)	Yes (*n* = 34)	HR (95% CI)	*p*-Value	HR (95% CI)	*p*-Value
Nitrotyrosine (nM)	98.39 ± 64.28	94.73 ± 60.97	107.79 ± 72.03	1.01 (1.00–1.01)	0.0090	1.01 (1.00–1.01)	0.0045
Protein carbonyl (pmol/mg)	0.42 ± 0.29	0.43 ± 0.29	0.39 ± 0.31	2.92 (0.94–9.11)	0.0652	1.53 (0.52–4.55)	0.4433
F2-isoprostanes (ng/mL)	47.35 ± 263.34	34.35 ± 168.92	80.64 ± 419.07				
log (F2-isoprostanes)	0.69 ± 0.94	0.76 ± 0.9	0.52 ± 1.03	0.86 (0.60–1.24)	0.4179		
4-HNE (µg/mL)	1.27 ± 3.5	1.39 ± 4.04	0.95 ± 1.41				
log (4-HNE)	−0.56 ± 1.02	−0.52 ± 1.01	−0.68 ± 1.03	0.95 (0.71–1.25)	0.7000		
MDA (µg/mL)	41.36 ± 30.97	39.02 ± 29.08	47.34 ± 35.04				
log (MDA)	1.52 ± 0.29	1.5 ± 0.27	1.56 ± 0.34	0.54 (0.17–1.68)	0.2866		
8-OHdG (ng/mL)	38.62 ± 28.99	35.28 ± 28.18	47.19 ± 29.61				
log (8-OHdG)	1.33 ± 0.79	1.26 ± 0.84	1.52 ± 0.63	1.59 (0.89–2.86)	0.1183		

4-HNE, hydroxynonenal; MDA, malondialdehyde; 8-OHdG, 8-hydroxy-2′-deoxyguanosine; AD, Alzheimer’s dementia; HR, hazard ratio; CI, confidence interval. Cox proportional hazards regression analyses were used for the association of oxidative stress markers with conversion to AD. The log10 transformation was applied to F2-isoprostanes, 4-HNE, MDA, and 8-OHdG values owing to their highly skewed distributions. * The covariates included in the multivariable analysis were identified using a univariable *p*-value threshold < 0.1, including age, education, depression onset age ≥ 75 years, number of depressive episodes, duration of the current episode, cardiac disease, and cerebrovascular disease.

## Data Availability

Data are contained within the article or Appendix A.

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
