# Peer review of "Oxidative Stress and Risk of Dementia in Older Patients with Depression: A Longitudinal Cohort Study Using Plasma Biomarkers"

_medicina, 2025, doi:10.3390/medicina61010108_

Round 1

Reviewer 1 Report

Comments and Suggestions for Authors

Jang et al. studied inflammatory markers in a cohort of patients with risk of dementia in patients with depression.

My comments are:

Write in the abstract how many years was the follow-up period.

Do the authors have the rights to use the data from the population studied?

Is the approval number unique for the current study?

Please check plagiarism report and lower for less than 20%.

Please, read instruction for authors for references description.

The rest of the manuscript I do not believe that needs changes.

Author Response

Reviewer #1:

Jang et al. studied inflammatory markers in a cohort of patients with risk of dementia in patients with depression.

My comments are:

Write in the abstract how many years was the follow-up period.

Do the authors have the rights to use the data from the population studied?

Is the approval number unique for the current study?

Please check plagiarism report and lower for less than 20%.

Please, read instruction for authors for references description.

The rest of the manuscript I do not believe that needs changes.

Comment 1: Write in the abstract how many years was the follow-up period.

>>> We appreciate your thoughtful comment regarding the inclusion of follow-up duration in the abstract. Based on your suggestion, we have included the follow-up period as a range (1.00–18.53 years) in the Abstract section of the manuscript to enhance clarity. Thank you for pointing this out.

[Abstract: Lines 31-32, page 1]

“During the follow-up period ranging from 1.00 to 18.53 years, 41 (28.08%) patients converted to AD.”

Comment 2: Do the authors have the rights to use the data from the population studied?

>>> We appreciate your comment regarding data usage rights. We confirm that the data utilized in this study were obtained in full compliance with the appropriate ethical guidelines and institutional policies. The study was approved by the Institutional Review Board of the Samsung Medical Center (approval no. 2006-03-012), and written informed consent was obtained from all participants prior to data collection. This information has been clarified in the Methods section. Thank you for highlighting this point.

Comment 3: Is the approval number unique for the current study?

>>> Thank you for your insightful question regarding the ethics approval number. We confirm that the approval number (IRB no. 2006-03-012) is specific to this study. This has been explicitly mentioned in the Methods section to ensure clarity and compliance with ethical standards. Thank you for considering these important details.

Comment 4: Please check plagiarism report and lower for less than 20%.

>>> We appreciate your comment regarding the plagiarism report. We have carefully reviewed the manuscript using a plagiarism detection tool to ensure its originality. The similarity index is now less than 20%, and any overlapping content has been revised or appropriately cited to ensure compliance with ethical publishing standards. Thank you for bringing this issue to our attention.

Comment 5: Please, read instruction for authors for references description.

>>> Thank you for your comment. We have reviewed the journal's citation policy and noted that some of our references may have violated these guidelines. Accordingly, we have revised the references to ensure full compliance with the policy. We appreciate your consideration of this matter, which has helped enhance the manuscript's adherence to the journal's standards.

Reviewer 2 Report

Comments and Suggestions for Authors  

The present study investigates the relationship between oxidative stress biomarkers and the risk of Alzheimer’s dementia (AD) in individuals with late-life depression. Among six biomarkers analyzed, nitrotyrosine emerged as a significant predictor of AD conversion, with a 5.14-fold increased risk. The findings suggest that oxidative stress, particularly protein nitration linked to reactive nitrogen species (RNS), may play a critical role in neurodegeneration in depressed populations.  The study is an original study within the scope of the journal. It needs some minor revisions.

1.     A specific research hypothesis should be added into introduction.  The rationale for the study and expected outcomes should be detailed in this section.

2.     More information on initial study and its purposes, sample should be detailed. The rationale for selecting this cohort (from a study on antidepressant treatment responses) should be explained.

3.     Justify several exclusion criteria. For instance the exclusion for psychiatric diagnosis’s are for lifetime diagnosis or current diagnosis. What was your criteria for "unstable psychiatric conditions“ to exclude them .

4.     Power analysis findings to justify the sufficiency of 146 participants for this study should be added to relevant section.

5.     The study seems to last over two decades. What was attrition rates? How many participants completed the follow-up? What have you done to address the variations in follow-up duration, how did you manage it in the analysis.

6.     Provide some brief information on measures used, who conducted then what were their competencies.  For the measures were they valid and reliable in used language?

7.     Why did you prefer to focus on the six biomarkers mentioned? Are these the most clinically relevant markers, or were there any other reason for selection?

8.     Some more limitations should be mentioned in text like not addressing potential confounding factors affecting biomarker levels (e.g., dietary habits, medication use, or smoking).

9.     Naturally, authors focused on the significant association of nitrotyrosine with AD risk. However they should also critically discuss why other oxidative stress markers did not show associations and what it means for their hypothesis.

10.   The discussion should also address potential contradictions in the literature regarding oxidative stress biomarkers and their utility in predicting dementia. Relevant inform and citations should be added and discussed.

Author Response

Reviewer #2:

The present study investigates the relationship between oxidative stress biomarkers and the risk of Alzheimer’s dementia (AD) in individuals with late-life depression. Among six biomarkers analyzed, nitrotyrosine emerged as a significant predictor of AD conversion, with a 5.14-fold increased risk. The findings suggest that oxidative stress, particularly protein nitration linked to reactive nitrogen species (RNS), may play a critical role in neurodegeneration in depressed populations.  The study is an original study within the scope of the journal. It needs some minor revisions.

  1. A specific research hypothesis should be added into introduction. The rationale for the study and expected outcomes should be detailed in this section.

  1. More information on initial study and its purposes, sample should be detailed. The rationale for selecting this cohort (from a study on antidepressant treatment responses) should be explained.

  1. Justify several exclusion criteria. For instance the exclusion for psychiatric diagnosis’s are for lifetime diagnosis or current diagnosis. What was your criteria for "unstable psychiatric conditions“ to exclude them .

  1. Power analysis findings to justify the sufficiency of 146 participants for this study should be added to relevant section.

  1. The study seems to last over two decades. What was attrition rates? How many participants completed the follow-up? What have you done to address the variations in follow-up duration, how did you manage it in the analysis.

  1. Provide some brief information on measures used, who conducted then what were their competencies. For the measures were they valid and reliable in used language?

  1. Why did you prefer to focus on the six biomarkers mentioned? Are these the most clinically relevant markers, or were there any other reason for selection?

  1. Some more limitations should be mentioned in text like not addressing potential confounding factors affecting biomarker levels (e.g., dietary habits, medication use, or smoking).

  1. Naturally, authors focused on the significant association of nitrotyrosine with AD risk. However they should also critically discuss why other oxidative stress markers did not show associations and what it means for their hypothesis.

  1. The discussion should also address potential contradictions in the literature regarding oxidative stress biomarkers and their utility in predicting dementia. Relevant inform and citations should be added and discussed.

Comment 1: A specific research hypothesis should be added into introduction.  The rationale for the study and expected outcomes should be detailed in this section.

>>> Thank you for your valuable suggestion to include a specific research hypothesis in the Introduction section. We agree that this addition will enhance clarity and focus of this study. Based on your recommendation, we have revised the Introduction section to include the following research hypotheses:

[Introduction: Lines 87-94, page 2]

To address this gap, we hypothesize that oxidative stress biomarkers are significantly associated with an increased risk of dementia subsequently developing in individuals with depression. Specifically, we propose that elevated levels of oxidative stress markers, such as ROS, and their downstream effects may serve as early indicators of dementia risk. By investigating the longitudinal relationship between oxidative stress biomarkers and dementia in a cohort of individuals with depression, this study aimed to clarify the predictive utility of these biomarkers and provide a foundation for personalized prevention strategies.” 

Comment 2: More information on initial study and its purposes, sample should be detailed. The rationale for selecting this cohort (from a study on antidepressant treatment responses) should be explained.

>>> We appreciate your comment and have added further details about the original cohort's population, initial purpose of the original study, and the rationale for selecting this population. Our initial cohort of individuals with depression consisted of those who met the Diagnostic and Statistical Manual of Mental Disorders, Fourth Edition (DSM-IV) criteria for major depressive disorder. This cohort comprised patients who initially sought treatment for depression and continued to receive care at the Samsung Medical Center. This naturalistic setting allowed many participants to be followed-up consistently, even beyond the initial studies on antidepressant treatment responses, facilitating the long-term tracking of clinical and biological changes. Over time, the cohort was expanded and followed-up longitudinally to explore various aspects of late-life depression, including clinical, genetic, pharmacokinetic, and immune factors. This robust dataset provided a unique opportunity to investigate the interplay between depression and dementia. For the current analysis, we selected a subset of participants aged 55 years or older who were cognitively normal at baseline and had available blood samples for biomarker analysis. This focus aligns with the aim of our study to examine the predictive utility of oxidative stress biomarkers in a high-risk population. The inclusion criteria ensured a comprehensive analysis of both clinical and biological data to address the research question. We have incorporated this explanation into the revised manuscript. 

[Materials and Methods: Lines 129-141, page 3]

This cohort comprised patients who initially sought treatment for depression and continued to receive care at the Samsung Medical Center. This naturalistic setting allowed many participants to be followed-up consistently, even beyond the initial studies on antidepressant treatment responses, facilitating the long-term tracking of clinical and biological changes. Over time, the cohort was expanded and followed-up longitudinally to explore various aspects of late-life depression, including clinical, genetic, pharmacokinetic, and immune factors. This robust dataset provided a unique opportunity to investigate the interplay between depression and dementia.

From this cohort, we identified 146 patients aged ≥ 55 years who were cognitively normal at baseline (Korean Mini-Mental State Examination [K-MMSE] score ≥ 28/30 [19]) and had available plasma samples for oxidative stress biomarker analysis. This subset allowed us to investigate the relationship between oxidative stress biomarkers and risk of dementia in cognitively healthy individuals with depression.”  

Comment 3: Justify several exclusion criteria. For instance the exclusion for psychiatric diagnosis’s are for lifetime diagnosis or current diagnosis. What was your criteria for "unstable psychiatric conditions” to exclude them .

>>> We appreciate the opportunity to clarify the exclusion criteria and have addressed these details in the revised manuscript. All psychiatric diagnoses for exclusion were assessed at baseline, meaning that they reflected the current diagnoses at the time of enrollment. Specifically, exclusions of those with psychotic disorders (e.g., schizophrenia or delusional disorder) and bipolar affective disorder were based on current diagnoses, as determined through clinical interviews using the DSM-IV criteria. These exclusions were made to ensure that the study population is focused on those with unipolar depression without significant comorbid psychiatric illnesses that may confound the relationship between depression and the risk of dementia. Unstable psychiatric conditions were defined as acute symptoms or events that posed immediate risks or challenges for stable follow-up. For example, patients who had recently attempted suicide during their current depressive episode were excluded. This criterion ensured the reliability of the baseline assessments and minimized the risk of attrition due to psychiatric instability during the study period. These clarifications have been incorporated into the revised manuscript to enhance transparency of the exclusion process. 

[Materials and Methods: Lines 96-128, pages 2-3]
“2.1. Study Cohort and Participant Selection

At baseline, patients were excluded if they had psychotic disorders (such as schizophrenia or delusional disorder), bipolar affective disorder, neurological illnesses (such as Par-kinson’s disease and epilepsy), intellectual developmental disability, significant medical conditions, history of alcohol or drug dependence, personality disorders, head trauma with loss of consciousness, malignancy, or abnormal baseline laboratory findings, to focus on unipolar depression and avoid the potential confounding effects of these comorbidities. Furthermore, patients with unstable psychiatric conditions posing immediate risks or challenges to stable follow-up (e.g., recent suicide attempts during a depressive episode) were also excluded. All participants were recruited from a geropsychiatric clinic at Samsung Medical Center between June 1998 and January 2012.” 

Comment 4: Power analysis findings to justify the sufficiency of 146 participants for this study should be added to relevant section.

>>> Thank you for your comment. As this study performed a retrospective analysis, an a priori power calculation was not feasible, which has been noted as a limitation in the Discussion section. In response to your concern regarding the sample size, we conducted a post hoc power calculation. For the adjusted hazard ratio of 1.01 for nitrotyrosine, the estimated post hoc power was 89%, reflecting robust statistical power, given an observed p-value of 0.0045. However, we acknowledge the inherent limitations of post hoc power calculations, including the potential to overestimate the effect size and their limited utility in validating study findings (1). Accordingly, we have revised the manuscript to include the post hoc power analysis findings while highlighting its limitations in the discussion section. 

[Discussion: Lines 566-570, pages 10]

Our post-hoc calculation showed that the power for the adjusted HR of 1.01 for nitro-tyrosine was 89%, indicating sufficient statistical power, given an observed p-value of 0.0045. While post-hoc power analyses are inherently limited and do not replace pre-study power calculations [43], this result indicates that our study had sufficient power to detect observed associations, thereby supporting the robustness of our findings.

Comment 5: The study seems to last over two decades. What was attrition rates? How many participants completed the follow-up? What have you done to address the variations in follow-up duration, how did you manage it in the analysis.

>>> Thank you for your thoughtful comment. This study employed a naturalistic design, where follow-up continued until participants experienced an endpoint, such as dementia diagnosis, death, or loss to follow-up. Consequently, calculating the traditional attrition rate was challenging. However, we tracked the participants throughout the study period, and the number of participants who completed the follow-up or were lost to follow-up is detailed in the revised Results section. To address variations in the follow-up duration, we used time-to-event analysis (e.g., Cox proportional hazards models), which accounts for differing observation times and ensures robustness in handling the variability inherent in naturalistic studies. We believe that this approach appropriately addresses the challenges posed by varying follow-up durations. Additionally, we incorporated this limitation into the revised Discussion section to acknowledge its potential impact and provide further context. Thank you for raising this important point, which has helped us clarify these details in the manuscript. 

[Results: Lines 322-325, page 5]

“Among the 146 participants, 41 (28.08%) converted to AD during the follow-up period ranging from 1.00 to 18.53 years, and median follow-up time (95% CI) was 6.2 (4.8-9.2) years estimated by reverse Kaplan–Meier method. Among the remaining 105 partici-pants, 34 died, 55 were lost to follow-up, and 16 completed the follow-up period by December 31, 2021.”

[Discussion: Lines 570-572, page 10]

“Third, the substantial dropout rate over the nearly 20-year follow-up period may have affected the internal validity of the study and reduced its statistical power.

Comment 6: Provide some brief information on measures used, who conducted then what were their competencies.  For the measures were they valid and reliable in used language?

>>> We appreciate the opportunity to elaborate on the measures used, who measured them, what their competencies were, as well as the validity and reliability of the tools employed. Psychiatric interviews were conducted by a board-certified psychiatrist specializing in geriatric psychiatry with over a decade of clinical experience. Cognitive functioning was assessed with the Korean Mini-Mental State Examination (K-MMSE) and other neuropsychological assessments, including the Clinical Dementia Rating (CDR) scale, the Seoul Neuropsychological Screening Battery-Dementia version (SNSB-D), Seoul-Activities of Daily Living (S-ADL), Seoul-Instrumental Activities of Daily Living (S-IADL), Korean version of the Neuropsychiatric Inventory (K-NPI), and the Korean version of the Geriatric Depression Scale (GDS). These tools have been validated and demonstrated strong reliability in Korean-speaking populations (2-6). Cognitive assessments were performed by clinical psychologists who had undergone specific training in the standardized administration of the tools used. Laboratory analyses of biomarkers were performed in a certified clinical laboratory by qualified biotechnologists with extensive experience in ELISA-based methods.  We acknowledge that while the references for the assessment tools have been included in the manuscript, by adding key details about their application and validity in the Materials and Methods section, clarity and accessibility of the study can be improved for the readers. This has been incorporated into the revised manuscript to reflect your suggestion. Thank you for emphasizing the importance of these aspects in ensuring the rigor and transparency of this study. 

[Materials and Methods: Lines 153-223, pages 3-4]

“A board-certified psychiatrist with extensive clinical experience in geriatric psychiatry confirmed all diagnoses using the SPES, clinical observations, and medical records.

The K-MMSE, a validated and reliable tool for cognitive screening in Korean-speaking populations [19], was updated annually.

These validated cognitive assessments were conducted by clinical psychologists trained in the standardized administration of these tools to ensure consistency and reliability of data collection.” 

Comment 7: Why did you prefer to focus on the six biomarkers mentioned? Are these the most clinically relevant markers, or were there any other reason for selection?

>>> We appreciate your insightful question regarding the selection of the six biomarkers. The decision to focus on nitrotyrosine, protein carbonyl, F2-isoprostanes, malondialdehyde (MDA), 4-hydroxynonenal (4-HNE), and 8-hydroxy-2'-deoxyguanosine (8-OHdG) was based on a comprehensive review of the literature (7-10). Several prior studies have highlighted that these biomarkers are commonly associated with oxidative stress and identified their roles in neurodegeneration and dementia. We also aimed to ensure that our selection of biomarkers is consistent across multiple studies, as indicators of oxidative stress in dementia and other related disorders. For instance, studies such as those by Baldeiras et al. (11) and Butterfield et al. (12) consistently propose nitrotyrosine and F2-isoprostanes as reliable biomarkers for tracking oxidative stress in Alzheimer’s disease and mild cognitive impairment. In addition to their clinical relevance, we prioritized biomarkers that were measurable and with high sensitivity and specificity using ELISA or similar methodologies to ensure the reliability of our data. By narrowing our focus to these six biomarkers, we sought to provide a comprehensive, yet targeted assessment of oxidative stress in relation to dementia risk in patients with late-life depression. We have incorporated these clarifications in the revised manuscript to enhance the transparency of our methodological approach. Thank you for your thoughtful comment. 

[Materials and Methods: Lines 239-242, page 4]

These six biomarkers were selected based on their consistent identification in previous studies as key indicators of oxidative stress linked to neurodegeneration and dementia, emphasizing their clinical relevance and suitability for measurements using established methodologies.

Comment 8: Some more limitations should be mentioned in text like not addressing potential confounding factors affecting biomarker levels (e.g., dietary habits, medication use, or smoking).

>>> Thank you for your insightful suggestion regarding additional limitations. We agree that unaccounted potential confounding factors, such as dietary habits, medication use, and smoking status may influence biomarker levels and therefore affect the study outcomes. We have incorporated the following text into the Discussion section to address these limitations: 

[Discussion: Lines 583-587, page 10]
Lastly, while we adjusted for various covariates in the analysis, other potential confounding factors, such as dietary habits, smoking status, and medication use, were not accounted for. These factors are known to influence oxidative stress biomarker levels and may have introduced bias into our findings.” 

Comment 9: Naturally, authors focused on the significant association of nitrotyrosine with AD risk. However they should also critically discuss why other oxidative stress markers did not show associations and what it means for their hypothesis.

>>> Thank you for your thoughtful comment. We critically discussed why other oxidative stress markers did not show significant associations, focusing on factors such as differences in biomarker stability, assay sensitivity, and the unique biological conditions of depression, including chronic HPA axis activation and elevated baseline oxidative stress levels. These factors may have homogenized the biomarker levels within the cohort, limiting the detection of significant differences. We hope that this addresses your concerns and enhances the clarity and rigor of the manuscript. 

[Discussion: Lines 500-513, page 9]

Other oxidative stress markers, such as F2-isoprostanes, 4-HNE, MDA, and 8-OHdG, did not show statistically significant associations, which contrasts with findings of previous general population-based studies [13,14,30,32]. This discrepancy could be attributed to several factors. First, differences in the biomarker stability and sensitivity of the assays used might have contributed to the variability in the measurements. Lipid peroxidation markers, such as MDA and 4-HNE, are highly reactive and may degrade during sample storage or handling, potentially affecting the reliability of their associations [37]. Second, the biological conditions of the individuals with depression might have influenced the results. Depression is often associated with chronic activation of the hypothalamic-pituitary-adrenal axis, which can lead to reduced antioxidant capacity and accumulation of oxidative stress [6]. This potential increase in oxidative stress levels among patients with depression may homogenize the biomarker levels within the cohort, thus reducing the ability to detect meaningful differences between individuals who develop AD and those who do not.”  

Comment 10: The discussion should also address potential contradictions in the literature regarding oxidative stress biomarkers and their utility in predicting dementia. Relevant inform and citations should be added and discussed.

>>> Thank you for your valuable comment. In response, we have expanded the Discussion section to address the potential contradictions in the literature regarding oxidative stress biomarkers and their utility in predicting dementia. Specifically, we included information on studies that found no significant association between nitrotyrosine levels and dementia, addressing potential methodological differences and sample size limitations that may explain these discrepancies. We also emphasized the importance of larger validation studies to establish the clinical utility of these biomarkers. The revised text documents these contradictions, while maintaining a balanced perspective on the current state of evidence. We have addressed this point in the Discussion section. 

[Discussion: Lines 477-499, pages 9-10]

“There are studies that did not find a significant association between nitrotyrosine levels and dementia. For example, one study reported no differences in plasma nitrotyrosine concentrations between patients with AD and age-matched controls [36]. This discrepancy may arise due to differences in study populations and methodological approaches used to measure the biomarkers. Additionally, the smaller sample size of that study (22 patients with AD and 18 age-matched controls) may have limited the power to detect significant associations. Further validation with larger sample sizes is essential to confirm the utility of nitrotyrosine in predicting dementia and to establish an optimal cutoff value for clinical use.

 References

  1. Heckman MG, Davis JM, 3rd, Crowson CS. Post Hoc Power Calculations: An Inappropriate Method for Interpreting the Findings of a Research Study. J Rheumatol. 2022;49(8):867-70. Epub 20220201. doi: 10.3899/jrheum.211115. PubMed PMID: 35105710.
  2. Ahn HJ, Chin J, Park A, Lee BH, Suh MK, Seo SW, et al. Seoul Neuropsychological Screening Battery-dementia version (SNSB-D): a useful tool for assessing and monitoring cognitive impairments in dementia patients. J Korean Med Sci. 2010;25(7):1071-6. Epub 20100617. doi: 10.3346/jkms.2010.25.7.1071. PubMed PMID: 20592901; PubMed Central PMCID: PMC2890886.
  3. Bae JN, Cho MJ. Development of the Korean version of the Geriatric Depression Scale and its short form among elderly psychiatric patients. J Psychosom Res. 2004;57(3):297-305. doi: 10.1016/j.jpsychores.2004.01.004. PubMed PMID: 15507257.
  4. Chin J, Park J, Yang SJ, Yeom J, Ahn Y, Baek MJ, et al. Re-standardization of the Korean-Instrumental Activities of Daily Living (K-IADL): Clinical Usefulness for Various Neurodegenerative Diseases. Dement Neurocogn Disord. 2018;17(1):11-22. Epub 20180331. doi: 10.12779/dnd.2018.17.1.11. PubMed PMID: 30906387; PubMed Central PMCID: PMC6427997.
  5. Kang HS, Ahn IS, Kim JH, Kim DK. Neuropsychiatric symptoms in korean patients with Alzheimer's disease: exploratory factor analysis and confirmatory factor analysis of the neuropsychiatric inventory. Dement Geriatr Cogn Disord. 2010;29(1):82-7. Epub 20100203. doi: 10.1159/000264629. PubMed PMID: 20130406.
  6. Kang Y ND, Hahn S. A validity study on the Korean Mini-Mental State Examination (K-MMSE) in dementia patients. J Korean Neurol Assoc. 1997;15:300–8.
  7. Collin F, Cheignon C, Hureau C. Oxidative stress as a biomarker for Alzheimer's disease. Biomark Med. 2018;12(3):201-3. Epub 20180213. doi: 10.2217/bmm-2017-0456. PubMed PMID: 29436240.
  8. Huang WJ, Zhang X, Chen WW. Role of oxidative stress in Alzheimer's disease. Biomedical reports. 2016;4(5):519-22.
  9. Ionescu-Tucker A, Cotman CW. Emerging roles of oxidative stress in brain aging and Alzheimer's disease. Neurobiol Aging. 2021;107:86-95. Epub 20210725. doi: 10.1016/j.neurobiolaging.2021.07.014. PubMed PMID: 34416493.
  10. Skoumalová A, Hort J. Blood markers of oxidative stress in Alzheimer's disease. J Cell Mol Med. 2012;16(10):2291-300. doi: 10.1111/j.1582-4934.2012.01585.x. PubMed PMID: 22564475; PubMed Central PMCID: PMC3823422.
  11. Baldeiras I, Santana I, Proença MT, Garrucho MH, Pascoal R, Rodrigues A, et al. Oxidative damage and progression to Alzheimer's disease in patients with mild cognitive impairment. J Alzheimers Dis. 2010;21(4):1165-77. doi: 10.3233/jad-2010-091723. PubMed PMID: 21504121.
  12. Butterfield DA, Reed TT, Perluigi M, De Marco C, Coccia R, Keller JN, et al. Elevated levels of 3-nitrotyrosine in brain from subjects with amnestic mild cognitive impairment: implications for the role of nitration in the progression of Alzheimer's disease. Brain Res. 2007;1148:243-8. Epub 20070307. doi: 10.1016/j.brainres.2007.02.084. PubMed PMID: 17395167; PubMed Central PMCID: PMC1934617.

Reviewer 3 Report

Comments and Suggestions for Authors

comment 1) the introduction is short and the authors did not describe the prevalence of dementia in older patients with depression. 

comment2)please give an outlook and impulse for further research. for example, the potential role of the gut microbiome in the risk of dementia in older patients with depression with a focus on oxidative stress and inflammation 

Comments on the Quality of English Language

check the manuscript for some minor errors 

Author Response

Reviewer #3:

comment 1) the introduction is short and the authors did not describe the prevalence of dementia in older patients with depression.

comment2) please give an outlook and impulse for further research. for example, the potential role of the gut microbiome in the risk of dementia in older patients with depression with a focus on oxidative stress and inflammation.

Comment 1: The introduction is short and the authors did not describe the prevalence of dementia in older patients with depression.

>>> Thank you for your thoughtful comment. In response, we have revised the Introduction section to provide a more comprehensive background. We included findings from meta-analyses and longitudinal studies, noting that older adults with depression have a significantly elevated risk of developing dementia with hazard ratios ranging from 1.85 to 2.83. This emphasizes the strong association between depression and dementia and underscores the need for the early identification of predictive biomarkers in this high-risk population. We also refined the research hypothesis to state that oxidative stress biomarkers are early indicators of dementia risk. These changes have enhanced the context and clarity of the objectives of the study. Thank you for your suggestions, which have strengthened the manuscript. 

[Introduction: Lines 57-94, page 2]

Meta-analyses and longitudinal studies have reported that older adults with depression have a significantly elevated risk of developing dementia, with hazard ratios ranging from 1.85 to 2.83 [2,3]. This strong association highlights the importance of identifying predictive biomarkers that can help detect early dementia risk in high-risk populations.…Previous studies have extensively examined the role of oxidative stress in dementia and have shown significantly higher levels of oxidative stress biomarkers in patients with Alzheimer’s disease than in controls [11]. However, these investigations have largely focused on the general population or on individuals already diagnosed with dementia [12], leaving a critical gap in the understanding of how oxidative stress contributes to the risk of dementia in individuals with depression. Furthermore, most previous research has been cross-sectional [13], limiting insights into the longitudinal association between oxidative stress biomarkers and the risk of dementia [14]. While previous studies [15] have explored the longitudinal relationship between oxidative stress and dementia progression, their follow-up duration was relatively short, ranging from 1 to 6 years. Given the growing evidence that depression is a prodrome or risk factor for dementia, it is imperative to identify specific biomarkers that can predict dementia in patients with de-pression.

To address this gap, we hypothesize that oxidative stress biomarkers are significantly associated with an increased risk of dementia subsequently developing in individuals with depression. Specifically, we propose that elevated levels of oxidative stress markers, such as ROS, and their downstream effects may serve as early indicators of dementia risk. By investigating the longitudinal relationship between oxidative stress biomarkers and dementia in a cohort of individuals with depression, this study aimed to clarify the predictive utility of these biomarkers and provide a foundation for personalized prevention strategies.” 

Comment 2: Please give an outlook and impulse for further research. For example, the potential role of the gut microbiome in the risk of dementia in older patients with depression with a focus on oxidative stress and inflammation.

>>> Thank you for your insightful suggestion. We have incorporated your recommendations by adding a future research direction in the revised manuscript. Specifically, we addressed the potential role of the gut microbiome in dementia risk among older adult patients with depression by focusing on its interactions with oxidative stress and inflammation. Recent studies exploring the relationship between depression and dementia and the gut microbiome have highlighted its important role in the regulation of oxidative stress and inflammation (13-15). These studies have revealed that alterations in the gut microbiome and its metabolites can exacerbate oxidative stress, influencing both depressive symptoms and the risk of dementia. Future research should focus on integrating gut microbiome profiles and oxidative stress biomarkers to develop innovative models that can predict the early transition to dementia in older adult patients with depression. Furthermore, it would be highly beneficial to clarify the relationship between the microbiome and oxidative stress markers and to evaluate the potential impact of microbiome-based therapies in preventing dementia. Such research could offer important biomarkers for assessing dementia risk in older adult patients with depression and contribute significantly to the development of personalized treatment approaches. We have incorporated the following text into the Discussion section to address these limitations:

[Discussion: Lines 537-561, pages 9-10]

Our findings provide a foundation for future research on the complex interplay between oxidative stress, inflammation, and neurodegeneration. Potential research might involve an investigation of the gut-brain axis, as recent studies have highlighted the influence of the gut microbiome on oxidative stress and inflammatory pathways [40-42]. Alterations in the composition of gut microbiota and their metabolites exacerbate systemic oxidative stress and inflammation, which are key contributors to depression and dementia. Integrating microbiome profiling with oxidative stress biomarkers may enhance predictive models and inform microbiome-targeted therapies to mitigate neuroinflammation and oxidative damage. Such research may lead to novel personalized approaches to prevent or delay dementia in at-risk populations.” 

References

  1. Cutuli D, Giacovazzo G, Decandia D, Coccurello R. Alzheimer's disease and depression in the elderly: A trajectory linking gut microbiota and serotonin signaling. Front Psychiatry. 2022;13:1010169. Epub 20221130. doi: 10.3389/fpsyt.2022.1010169. PubMed PMID: 36532180; PubMed Central PMCID: PMC9750201.
  2. Luca M, Di Mauro M, Di Mauro M, Luca A. Gut Microbiota in Alzheimer's Disease, Depression, and Type 2 Diabetes Mellitus: The Role of Oxidative Stress. Oxid Med Cell Longev. 2019;2019:4730539. Epub 20190417. doi: 10.1155/2019/4730539. PubMed PMID: 31178961; PubMed Central PMCID: PMC6501164.
  3. Sochocka M, Donskow-Łysoniewska K, Diniz BS, Kurpas D, Brzozowska E, Leszek J. The Gut Microbiome Alterations and Inflammation-Driven Pathogenesis of Alzheimer's Disease-a Critical Review. Mol Neurobiol. 2019;56(3):1841-51. Epub 20180623. doi: 10.1007/s12035-018-1188-4. PubMed PMID: 29936690; PubMed Central PMCID: PMC6394610.

Round 2

Reviewer 2 Report

Comments and Suggestions for Authors

Auhtors responded my comments adequately.